# Overexpression of the Apple (*Malus*
*× domestica*) *MdERF100* in *Arabidopsis* Increases Resistance to Powdery Mildew

**DOI:** 10.3390/ijms22115713

**Published:** 2021-05-27

**Authors:** Yiping Zhang, Li Zhang, Hai Ma, Yichu Zhang, Xiuming Zhang, Miaomiao Ji, Steve van Nocker, Bilal Ahmad, Zhengyang Zhao, Xiping Wang, Hua Gao

**Affiliations:** 1State Key Laboratory of Crop Stress Biology in Arid Areas, College of Horticulture, Northwest A&F University, Xianyang 712100, China; zyp9709@nwafu.edu.cn (Y.Z.); zhangli0221@nwafu.edu.cn (L.Z.); mahai@nwafu.edu.cn (H.M.); jyl74@nwsuaf.edu.cn (Y.Z.); zhangxiuming@nwafu.edu.cn (X.Z.); jimiaomiao@nwafu.edu.cn (M.J.); bajwa1999@nwafu.edu.cn (B.A.); zhaozy@nwsuaf.edu.cn (Z.Z.); 2Key Laboratory of Horticultural Plant Biology and Germplasm Innovation in Northwest China, Ministry of Agriculture, Northwest A&F University, Xianyang 712100, China; 3Department of Horticulture, Michigan State University, East Lansing, MI 48824, USA; vannocke@msu.edu

**Keywords:** apple, *MdERF100*, *MdbHLH92*, powdery mildew, disease resistance

## Abstract

APETALA2/ETHYLENE RESPONSIVE FACTOR (AP2/ERF) transcription factors play important roles in plant development and stress response. Although *AP2/ERF* genes have been extensively investigated in model plants such as *Arabidopsis thaliana*, little is known about their role in biotic stress response in perennial fruit tree crops such as apple (*Malus × domestica*). Here, we investigated the role of *MdERF100* in powdery mildew resistance in apple. MdERF100 localized to the nucleus but showed no transcriptional activation activity. The heterologous expression of *MdERF100* in *Arabidopsis* not only enhanced powdery mildew resistance but also increased reactive oxygen species (ROS) accumulation and cell death. Furthermore, *MdERF100*-overexpressing *Arabidopsis* plants exhibited differential expressions of genes involved in jasmonic acid (JA) and salicylic acid (SA) signaling when infected with the powdery mildew pathogen. Additionally, yeast two-hybrid and bimolecular fluorescence complementation assays confirmed that MdERF100 physically interacts with the basic helix–loop–helix (bHLH) protein MdbHLH92. These results suggest that *MdERF100* mediates powdery mildew resistance by regulating the JA and SA signaling pathways, and *MdbHLH92* is involved in plant defense against powdery mildew. Overall, this study enhances our understanding of the role of *MdERF* genes in disease resistance, and provides novel insights into the molecular mechanisms of powdery mildew resistance in apple.

## 1. Introduction

Apple (*Malus × domestica*) is one of the most widely cultivated and economically valuable fruit crops worldwide [1]. In China, improvements in apple production technology have led to its cultivation in new areas. However, apple production is constantly challenged by biotic and abiotic pressures. Apple powdery mildew, caused by the obligate vegetative fungus *Podosphaera leucotricha*, is one of the most ubiquitous and devastating diseases in apple worldwide [2]. This fungus damages various tissues and organs of apple trees, including new buds, leaves, flowers and young fruits. Powdery mildew symptoms first appear on the leaves in the form of leaf curling, appearance of white powder on the leaf blade, drying of leaf tips and leaf abscission [3]. Apple producers commonly use fungicides to reduce the losses caused by powdery mildew, but the continuous use of fungicides has negative impacts on fruit production and the environment [4]. Therefore, it is essential to characterize the molecular basis of powdery mildew resistance and to develop powdery mildew-resistant cultivars.

The *APETALA2/ETHYLENE RESPONSE FACTOR* (*AP2/ERF*) genes encode plant-specific transcription factors, which perform important functions in plant development and response to various stresses, including pathogen infection, salt, wounding, drought, hypoxia and high temperatures [5,6]. The *ERF* genes form the largest subfamily of the *AP2/ERF* gene family and contain an AP2 domain [7]. *ERF* genes have been well studied in the model plant *Arabidopsis thaliana* in the context of resistance to various pathogens, including *Ralstonia solanacearum*, *Alternaria alternata*, tobacco mosaic virus (TMV), *Botyosphaeria dothidea*, and *Pseudomonas syringae* pv. tomato DC3000 (PstDC3000). These genes have also been studies in other plant species. For example, heterologous expression of the cotton (*Gossypium hirsutum*) *ERF* gene *GhB301* in *Nicotiana benthamiana* enhanced its resistance against fusarium wilt [8]. The basic helix–loop–helix (bHLH) superfamily is considered as the second largest class of plant transcription factors [9]. A bHLH domain contains 50–60 amino acids and two functionally distinct regions: the N-terminal basic region, which is involved in the binding to the E-box (CANNTG) DNA sequence, and the HLH domain, which mediates domain dimerization [10,11]. Numerous investigations have shown that bHLH transcription factors play important roles in abiotic stresses, including cold [12], drought [13], salinity [14] and iron deficiency [15]. Moreover, it has been shown that bHLHs participate in plant defense against pests and pathogens, such as *Xanthomonas euvesicatoria* [16]. Some pathogensis-related (*PR*) genes, involved in salicylic acid (SA) signaling, were downregulated in the *OsHLH61* RNA interference (RNAi) plants [17], indicating a close relationship between bHLHs and the SA signaling pathway.

Plant hormones such as SA and jasmonic acid (JA) play crucial roles in various defensive responses [18]. In *Arabidopsis*, the *ENHANCED DISEASE SUSCEPTIBILITY 1* (*EDS1*) gene encodes a lipase-like nucleo-cytoplasmic protein that regulates SA accumulation [19,20]. The *NONEXPRESSOR OF PR1* (*NPR1*) gene encodes an SA receptor, which is critical for SA-mediated defense responses and pathogen resistance [19,21]. In *Arabidopsis*, PR proteins encoded by *AtPR2*, *AtPR3* and *AtPR5* genes are involved in SA and JA regulation [22,23,24]. The defense gene *AtPDF1.2* also plays an important role in JA-mediated defense pathways [25].

Thus far, our understanding of the defense molecular mechanisms of powdery mildew is limited. In grapevine, the expression of the *VqSTS6* gene from *Vitis quinquangularis* enhanced powdery mildew resistance by increasing the content of stilbenes [8]. The ectopic expression of *VvDOF3* in *Arabidopsis* conferred enhanced resistance to powdery mildew through the SA signaling pathway [26]. The overexpressing of *ERF1-V* in wheat (*Triticum aestivum*) could improve resistance to powdery mildew [27]. Meanwhile, only a limited number of studies have addressed the molecular mechanisms of powdery mildew susceptibility and resistance in apple. The knockdown of the susceptibility (*S*) gene *MdMLO19* in apple resulted in enhanced resistance against powdery mildew [4]. Similarly, transgenic apple plants expressing the *MhNPR1* gene from *Malus hupehensis* also showed enhanced resistance against powdery mildew [28]. Furthermore, there have been no reports on the potential role of *ERF* genes in powdery mildew resistance in apple. According to the previous study, *MdERF100* was a differentially expressed gene selected from the transcriptome data of apple inoculated with *Podosphaera leucotricha* [1]. We suspect that *MdERF100* functions as a positive regulator in powdery mildew resistance. To confirm our hypothesis, we performed a heterologous expression analysis of *MdERF100* in *Arabidopsis.* In a futher study, we showed that MdERF100 physically interacts with MdbHLH92, and plays a positive role in SA-mediated defense against powdery mildew. The research aims to provide candidate genes for the development of powdery mildew-resistant apple cultivars, and lay the foundation for the investigation of the molecular mechanisms of disease resistance in apple.

## 2. Results

### 2.1. Expression Pattern and Characterization of MdERF100

To evaluate the potential role of *MdERF100* in disease resistance in apple, we analyzed its expression in the leaves of apple seedlings inoculated with *Podosphaera leucotricha*. The results showed that the expression of *MdERF100* was strongly upregulated at 6 h post-inoculation (hpi), maintained at least until 12 hpi, downregulated at 24 and 48 hpi and significantly upregulated at 72 hpi (Figure 1a). To determine whether MdERF100 responds to hormone signaling pathways, we treated apple leaves with SA, JA and ethephon (Eth), and monitored *MdERF100* expression by quantitative real-time PCR (qRT-PCR). Treatment with all three hormones significantly upregulated the expression of *MdERF100* (Figure 1b–d). Compared with the control, the expression of *MdERF100* was up to 25.1-fold higher in the SA treatment at 12 hpi (Figure 1b) and 8.6- and 3.9-fold higher in the JA and Eth treatments, respectively, at 3 hpi (Figure 1c,d). The results suggest that MdERF100 functions as an important regulator in apple powdery mildew defense via the SA, JA, and Eth signaling pathways.

The *MdERF100* gene is located on chromosome 06 on the apple genome (Apple Genome Browser: https://www.rosaceae.org, accessed on 10 March 2020). The coding sequence (CDS) of *MdERF100* is 831 bp in length and is predicted to encode a 276-amino acid (aa) protein, with a molecular weight of ~29.6 kDa and an isoelectric point (pI) of 8.45 (Figure 1e). The MdERF100 protein contains a highly conserved AP2 domain (142–206 aa), as shown by the multiple sequence alignment of ERF proteins (Figure 1f). Additionally, phylogenetic analysis showed that MdERF100 is most closely related to PbERF2-like and PpERF1A (Figure 1g).

### 2.2. MdERF100 Shows Nuclear Localization but No Transcriptional Activation Activity

To determine the subcellular location of MdERF100, the open reading frame (ORF) of *MdERF100* was subcloned downstream of the cauliflower mosaic virus (CaMV) 35S promoter with the green fluorescent protein (*GFP*), and the resulting *35S:MdERF100-GFP* was transiently transformed into *Nicotiana benthamiana* leaf epidermal cells (Figure 2a). Leaf sections around the injection site were analyzed by confocal microscopy at approximately 24 h post-infiltration. The green fluorescence signal was constrained to several discrete foci corresponding to the position of nuclei, revealing that the MdERF100-GFP fusion localized to the nucleus (Figure 2a). To determine the transcriptional activation potential of MdERF100, we performed yeast two-hybrid (Y2H) assays. All yeast cells grew well on synthetic-defined medium lacking tryptophan (SD/-Trp). Yeast cells transformed with pGBKT7-MdERF100, empty vector (pGBKT7) or negative control (co-transformation of pGBKT7-Lam with pGADT7-T) did not survive on the SD medium lacking Trp, X-α-Gal and Aureobasidin A (SD/-Trp/X-α-Gal/AbA), while those transformed with the positive control (co-transformation of pGBKT7-53 with pGADT7-T) produced blue colonies, indicating that MdERF100 exhibited no activation ability in yeast (Figure 2b).

### 2.3. Heterologous Expression of MdERF100 in Arabidopsis Confers Powdery Mildew Resistance

To determine whether MdERF100 promotes disease resistance when expressed heterologously in *Arabidopsis*, we engineered transgenic *Arabidopsis* lines expressing *MdERF100* under the control of the strong, constitutive CaMV 35S promoter (*35S:MdERF100*), and evaluated their resistance to powdery mildew. Three transgenic lines significantly overexpressing *MdERF100* compared with wild-type (WT) were selected for further experiments (Appendix A). The *35S:MdERF100* transgenic lines showed enhanced resistance to powdery mildew compared with WT control plants (Figure 3a). All three *35S:MdERF100* transgenic lines showed significantly fewer and less severe symptoms, as indicated by the disease severity index. In addition, significantly fewer spores were recovered from the leaves of transgenic lines than from those of WT (Figure 3b). Furthermore, we stained the leaves of transgenic and WT plants with trypan blue, diaminobenzidine (DAB) and nitroblue tetrazolium (NBT) to visualize dead cells and the accumulation of reactive oxygen species (ROS) (Figure 3c). The results showed that the number of dead cells and the level of ROS were higher in transgenic lines than in the WT controls. These results indicate that powdery mildew resistance is accompanied by a strong allergic and lethal reaction because of ROS burst after invasion by the fungal pathogen.

### 2.4. Expression Analysis of Disease Resistance (R) Genes in MdERF100-Overexpressing Plants in Response to E. necator Inoculation

To understand the mechanism of enhanced powdery mildew resistance in *MdERF100*-overexpressing plants, we analyzed the relative expression levels of key genes involved in hormone biosynthesis or signal transduction pathways. These included genes involved in SA signaling (*AtEDS1*, *AtPR2*, *AtPR5* and *AtNPR1*) and JA signaling (*AtPDF1.2* and *AtPR3*) (Figure 4). The expression of these genes was examined by quantitative PCR (qPCR) in 4-week-old seedlings of all three *35S:MdERF100* transgenic lines at 0, 24, 72 and 120 hpi. The expression of *AtEDS1* was higher at 24 and 72 hpi compared with the inoculated control (WT) plants, and was less than those in the control plants by 120 hpi (Figure 4a). Both *AtNPR1* and *AtPR2* were significantly upregulated at 24, 72 and 120 hpi in all three transgenic lines (Figure 4b,c). In addition, *AtPR5* showed a significantly greater expression at 24 and 72 hpi in all three transgenic lines, and relatively less expression at 120 hpi in two transgenic lines (Figure 4f). However, both *AtPDF1.2* and *AtPR3* were downregulated at 24, 72 and 120 hpi in transgenic lines compared with WT plants (Figure 4d,e). These results suggest that *MdERF100* positively regulates resistance to powdery mildew via the SA signaling pathway but negatively regulates powdery mildew resistance through the JA signaling pathway.

### 2.5. MdERF100 Physically Interacts with MdbHLH92

We used the Y2H system to test for interactions between MdERF100 and five putative interacting partners, including MdCSC1, MdBIP, MdbHLH92, MdPAE1 and MdUBQ, which were obtained from the STRING database, and genes encoding these proteins were cloned from the apple cultivar Gala. *MdERF100* was fused to the DNA-binding domain (BD) in the pGBKT7 (Clonetech, Palo Alto, CA, USA) vector, while genes encoding the putative interacting partners were fused to the activation domain (AD) in the pGADT7 (Clonetech, Palo Alto, CA, USA) vector. The *MdERF100* construct was co-transformed with each of the AD constructs into yeast cells. The co-transformed yeast cells cultured on SD/-Trp/-Leu (double dropout (DDO) medium) produced white colonies, whereas only yeast cells co-transformed with *MdbHLH92* and cultured on SD/-Ade-His-Leu-Trp (quadruple dropout (QDO) medium) with X-α-Gal and AbA produced blue colonies (Figure 5a). Moreover, as shown in Figure 5b, yellow fluorescent signals were observed in protoplasts co-expressing *MdERF100* and *MdbHLH92*. However, no fluorescent signal was detected in protoplasts transformed with other plasmid combinations (Figure 5b), indicating that MdERF100 interacted specifically with MdbHLH92 in yeast cells and *Arabidopsis* protoplasts (Figure 5).

### 2.6. Isolation of MdbHLH92 and Its Response to Powdery Mildew 

We performed qRT-PCR analysis to determine whether the expression of *MdbHLH92* cloned from the apple cultivar Gala was affected by powdery mildew infection. Compared with the mock control, the expression level of *MdbHLH92* was significantly upregulated at 48, 72 and 96 hpi, reaching a peak at 48 hpi (Figure 6a). The *MdbHLH92* gene, located on chromosome 16, has a CDS of 708 bp, and the MdbHLH92 protein harbors a 49 aa HLH domain (Figure 6b). The molecular weight and theoretical pI of MdbHLH92 are ~27.4 kDa and 9.28, respectively. Furthermore, MdbHLH92 is the most closely related to PbbHLH92-like (Figure 6c). Additionally, MdbHLH92 localized to the cytomembrane (Figure 6d) and showed no transcriptional activation ability in yeast cells (Figure 6e).

## 3. Discussion

Apple is a popular and economically important fruit, but apple trees often encounter various biotic stresses, such as powdery mildew infection, during their life time. According to previous studies, ERF transcription factors respond to various abiotic stresses, although their role in defense against biotic stresses should be given more importance. For instance, the AP2/ERF transcription factor TINY not only promotes drought response, but also suppresses brassinosteroid (BR)-regulated growth in *Arabidopsis* [29]. In rice (*Oryza sativa*), both drought tolerance and ethylene emission were reduced in *OsERF3*-overexpressing transgenic plants compared with the control with inhibited expression of *OsERF3*, indicating that *OsERF3* plays an important role in regulating ethylene biosynthesis and stress response [30]. Studies have shown that MdERF38 plays key roles in inducing anthocyanin biosynthesis under drought stress [31], while MdERF17 is involved in iron deficiency response [32]. However, the disease resistance function of ERF transcription factors in apple need to be explored.

In this study, qRT-PCR analysis showed that, after being inoculated with *Podosphaera leucotricha**,* the expression of *MdERF100* was significantly higher at 6, 12 and 72 hpi, but lower at 24 and 48 hpi compared with the control group inoculated with sterile water (Figure 1a). The expression level of *MdERF100* in Gala plants changed significantly after inoculation with the powdery mildew pathogen, indicating a close relationship between MdERF100 and the powdery mildew resistance mechanism. These results suggest that the *MdERF100* confers apple plants with the ability to respond to powdery mildew infection. Furthermore, *MdERF100* expression was significantly upregulated after SA, JA, and Eth hormone treatments, indicating that MdERF100 participates in these hormone signaling pathways. Amino acid sequence analyses showed that MdERF100 contains a putative AP2 domain. In soybean (*Glycine max*), GmERF75 has been shown to localize to the nucleus [33]. Similarly, in *Catharanthus roseus*, CrERF5 localizes to the nucleus and exhibits transcriptional activation activity [34]. These studies suggest that ERF function as transcriptional activators. In this study, although MdERF100 localized to the nucleus, it did not show transcriptional self-activation in yeast cells. It is possible that MdERF100 requires a post-translational modification or assistance from other proteins to regulate the expression of downstream genes.

In studies of *MdERF11* it was shown to enhance resistance to apple ring rot (*Botryosphaeria dothidea*) via SA signaling pathway [35], and in those of *MdERF3* it was shown to respond to *Botrytis cinerea* [36], suggesting that ERFs in apple might enhance defense responses against fungal diseases. Moreover, six *ERF* genes (*VpERF1*, *VpERF2*, *VpERF3*, *VqERF112*, *VqERF114* and *VqERF72)* from *Vitis pseudoreticulata* and *Vitis quinquangularis* showed responses to powdery mildew in different expression patterns [37,38]. These reports showed ERFs have an important function in regulating the defensive response to powdery mildew. In the current study, three transgenic *Arabidopsis* lines overexpressing *MdERF100* showed an enhanced defense response to powdery mildew compared with WT plants. Additionally, the transgenic lines showed a lower conidiophore count than WT plants at 7 days post-inoculation (dpi). A higher number of dead cells and increased ROS accumulation in transgenic lines suggest that *MdERF100* acts as a positive regulator of powdery mildew resistance. The result is consistent with previous studies and makes our findings related to *MdERF100* more reliable. Furthermore, hormonal signaling pathways also play an important role in the regulation of disease resistance mechanisms in plants, and the cross-communication between these pathways is antagonistic or synergistic [22,39]. For example, AtERF014 acts as a positive regulator of the SA signaling pathway and a negative regulator of the JA signaling pathway in *Arabidopsis* plants inoculated with PstDC3000 [40]. Previous researchers investigated the gene participating pathways by measuring the expression of marker genes in hormonal signaling pathways [35,41]. The overexpression of *MhNPR1* in *Nicotiana benthamiana* improved the expression of *PR* genes related to the SA signaling pathway, and could enhance broad-spectrum resistance to apple cultivar pathogens [41]. Moreover, the overexpression of *VdGATA2* in *Arabidopsis* can significantly improve the resistance to powdery mildew through increasing the expression of the key genes in the SA signaling pathway (*PR1*), which indicates that the SA signaling pathway is significant in powdery mildew resistance [42]. In our study, qRT-PCR analysis showed that the expression of a key SA signaling gene (*PR1*) was high, while that of a key gene involved in JA signaling (*PDF1.2*) was low (Figure 7). Taken together, these data support the hypothesis that MdERF100 regulates powdery mildew resistance by activating SA and suppressing JA signaling pathways.

Previous studies showed that the bHLH transcription factors regulate plant defense against a variety of abiotic and biotic stresses by binding to the E-box *cis*-acting element in the promoters of stress-responsive genes, thus controlling their expression [43]. In this study, the interaction of MdbHLH92, isolated from Gala, with MdERF100 was confirmed by Y2H and BiFC assays, suggesting that MdbHLH92 most likely participates in powdery mildew resistance (Figure 7). Whereas MdbHLH92 localized to the cytomembrane, AtbHLH92 localizes to the nucleus [44]. Moreover, we demonstrated the interaction between MdERF100 and MdbHLH92 in the nucleus of *Arabidopsis* protoplast, which implies that protein activation and migration possibly occurred before the interaction. Furthermore, the expression of *MdbHLH92* was significantly upregulated at 48 hpi, indicating that *MdbHLH92* responds to powdery mildew infection. Further investigations should be carried out to verify the function of MdbHLH92 in powdery mildew resistance through heterologous expression analysis in *Arabidopsis* and homologous expression assays in apple. Moreover, two bHLH transcription factors (TSAR1 and TSAR2) were reported to regulate the biosynthesis of triterpene saponins, which is confirmed to respond to a variety of fungal pathogens [45]. Overexpressing *GhbHLH171* in cotton improves its tolerance to the fungus *Verticillium dahliae* [46]. Although some research provides evidence of the defense function of bHLHs against fungal diseases, our understanding is limited. Previous studies have showed that some bHLH proteins, such as bHLH3 (AT4G16430), bHLH13 (AT1G01260), bHLH14 (AT4G00870) and bHLH17 (AT2G46510/AtAIB), function as negative regulators of JA signaling [46,47], whereas others, such as OsHLH61, act as positive regulators of SA signaling [17]. Coincidentally, our results showed that MdERF100 functions as a positive SA signaling regulator and negative JA signaling regulator, which is supported by the results of our expression analysis, thus providing new insights into the molecular mechanisms underlying the regulation of powdery mildew defense response in apple (Figure 7).

Overall, we showed that MdERF100 acts as a transcription factor and plays a positive regulatory role in enhancing powdery mildew resistance through the SA and JA signaling pathways. Furthermore, we showed that MdERF100 and MdbHLH92 directly interact with each other, which provides a new angle for investigating the transcriptional networks regulated by *MdERF100* in response to apple powdery mildew.

## 4. Materials and Methods

### 4.1. Plant Materials and Treatments

Apple (*Malus × domestica* cv. Gala) trees were maintained under natural environmental conditions at the White Water Apple Test Station of Northwest A&F University, Shaanxi, China. Young leaves still attached to the tree were inoculated with a culture of the laboratory strain of *Podosphaera leucotricha* containing 1 × 10^6^ spores/mL. Leaves of potted Gala plants grown in the greenhouse were treated with 5 mM SA, 50 μM methyl jasmonate (MeJA) and 5 mM Eth [48]. Samples were collected at 0, 3, 6, 12 and 24 h after the hormone treatments. Leaves sprayed with sterile water were used as a control. *Arabidopsis thaliana* ecotype Columbia (Col-0; WT) plants were grown in a culture chamber under a 16 h light/8 h dark photoperiod and 70% relative humidity. Six- to eight-week-old *Arabidopsis* plants were subjected to *Agrobacterium*-mediated transformation to generate stable transgenic lines. Transient transformation of *Nicotiana benthamiana* (line NC89) plants was performed by agroinfiltration.

### 4.2. Bioinformatics Analysis

The chromosomal location of *MdERF100* and *MdbHLH92* was predicted by Blat-Search in the Apple Genome Browser. Their conserved protein domains were identified using the Simple Modular Architecture Research Tool (SMART; http://smart.embl-heidelberg.de/, accessed on 10 September 2020). The PROSITE database (http://www.expasy.org/tools/, accessed on 10 September 2020) was used to characterize the physical and chemical properties of MdERF100 and MdbHLH92, and the ProtParam program (http://web.expasy.org/protparam/, accessed on 10 September 2020) was used to predict molecular mass. Ten homologous genes of *MdERF100* and thirteen of *MdbHLH92* were downloaded from NCBI (https://www.ncbi.nlm.nih.gov, accessed on 20 September 2020). The multiple sequence alignment of *MdERF100* was generated using the DNAMAN software. The MEGA 7.0 software [49] was used to build a phylogenetic tree of the two genes and their homologous genes with the Neighbor Joining (NJ) method and 1000 bootstrap replicates, respectively.

### 4.3. Subcellular Localization and Transcriptional Activation Assays

The CDSs of *MdERF100* and *MdbHLH92* were amplified from Gala plants using primers designed by Primer Premier 5.0 (Appendix A) in a system containing 100 ng each of the forward and reverse primers, 500 ng of cDNA template and 1 U of PCR Master Mix. The amplified fragment was inserted into the pCAMBIA2300-GFP vector, a reconstructed vector from pCAMBIA2300 (Cambia, Brisbane, QLD, Australia), and the resulting constructs (*MdERF100-GFP* and *MdbHLH92-GFP*) were introduced into *Agrobacterium tumefaciens* strain GV3101. *Nicotiana benthamiana* plants were agroinfiltrated and grown at 22 °C under 60% relative humidity and a 16 h light/8 h dark photoperiod. GFP signals were observed with a confocal microscope (Olympus FV1000, Tokyo, Japan). The CDSs of *MdERF100* and *MdbHLH92* were cloned into the pGBKT7 vector, which contains the DNA-binding region of GAL4, using *Eco*RI and *Bam*HI restriction endonucleases, as described previously [50]. The recombinant vector was transformed into yeast strain Y2H Gold using the Matchmaker^®^ Gold Yeast Two Hybrid System (Takara, Tokyo, Japan), according to the manufacturer’s instructions. The co-transformations of pGBKT7-53 with pGADT7-T and of pGBKT7-Lam with pGADT7-T into yeast cells were used as positive and negative controls, respectively. The transformants were grown at 30 °C for 3 days. Transcriptional activation was determined by blue colonies on selective medium (SD-Trp) supplemented with 40 μg/mL X-α-Gal and 200 ng/mL AbA. The primers used for plasmid construction are listed in Appendix A.

### 4.4. Inoculation of Arabidopsis Plants with Powdery Mildew Pathogen

The *MdERF100* CDS was cloned downstream of the CaMV 35S promoter in the plant expression vector pCAMBIA2300-GFP, and the resultant construct was introduced into *A. tumefaciens* strain GV3101. The floral dip method [51] was used to transform *Arabidopsis* seedlings. Seeds of T0 plants were selected on Murashige and Skoog (MS) medium supplemented with 50 mg/L kanamycin. Homozygous T3 lines were selected from three independent (T1) lines, and the homozygous T3 plants were used for subsequent experiments. T3 plants with five to seven leaves were inoculated with *Golovinomyces cichoracearum* UCSC1, an obligate parasitic powdery mildew pathogen of *Arabidopsis* stored on the leaves of WT plant, through rubbing the healthy leaves with infected leaves gently. WT plants were used as the control. Samples were collected before inoculation (0 h) and at 24, 72 and 120 hpi. All samples were immediately frozen in liquid nitrogen and stored at −80 °C. To perform trypan blue, DAB, and NBT staining, samples were collected at 168 hpi (7 dpi).

### 4.5. RNA Isolation and Gene Expression Analysis

Total RNA was extracted from using the E.Z.N.A. Plant RNA Kit (R6827-01, Omega Bio-tek, Doraville, GA, USA). The reverse transcription reaction was performed in two steps with the *Evo M-MLV* RT Kit with gDNA Clean for qPCRII (AG11711, Accurate Biotechnology, Changsha, Hunan, China), referring to the manufacturer’s instructions. Then, the synthesized cDNA was diluted six-fold to use as a template in qRT-PCR, which was performed using ChamQTM SYBR^®^ qPCR Master Mix (Vazyme) for fluorescence quantitation in a reaction system of 20 µL (Appendix A) under the following conditions: 95 °C for 30 s, followed by 42 cycles at 95 °C for 10 s and 60 °C for 30 s. Relative expression levels were analyzed with the 2^-ΔΔCT^ method. Graphs were generated with SigmaPlot 12.0 (Systat Software, San Jose, CA, USA) using relative expression values.

### 4.6. Y2H and BiFC Assays

Ten proteins were predicted to interact with MdERF100 using a Search Tool for the Retrieval of Interacting Genes/Proteins (STRING; https://string-db.org/cgi/input, accessed on 25 May 2020). After confirming the CDSs of genes encoding these proteins with the NCBI database (http://blast.ncbi.nlm.nih.gov/Blast.cgi, accessed on 25 May 2020), gene-specific primers containing *Eco*RI and *Bam*HI restriction sites were designed with Primer Premier 5.0 (Appendix A). Then, the CDSs of five genes (*MdCSC1*, *MdBIP*, *MdbHLH92*, *MdPAE1* and *MdUBQ*) were cloned successfully from Gala and inserted into the pGADT7 vector. The resulting vectors were co-transformed into the yeast strain Y2H and cultivated on SD/-Trp/-Leu (DDO), SD/-Ade-His-Leu-Trp (QDO) and QDO supplemented with 40 μg/mL X-α-Gal and 200 ng/mL AbA (QDO/X/A). The growth of yeast colonies was observed after 3–4 days; blue colonies on QDO/X/A indicated protein–protein interaction. Full-length CDSs of *MdERF100* and *MdbHLH92* were cloned into pUCSPY-CE and pUCSPY-NE vectors [52] using *Xho*I and *Kpn*I restriction enzymes. The recombinant plasmids were co-transformed into *Arabidopsis* protoplasts as described previously [53]. After 22 h, yellow fluorescent protein (YFP) signals were visualized by confocal laser scanning microscope (LEICA TCS SP8, Leica, Bensheim, Germany). The primers used for plasmid construction are listed in Appendix A.

## 5. Conclusions

Consistent with its presumed identity as a transcriptional regulator, MdERF100 localized to the nucleus and did not activate gene transcription. The overexpression of *MdERF100* in *Arabidopsis* enhanced resistance to powdery mildew and influenced the expression of genes involved in SA and JA signaling pathways. Additionally, MdERF100 showed direct physical interaction with MdbHLH92 in Y2H and BiFC assays. Thus, this study provides preliminary information about the function of *MdERF100* in powdery mildew resistance.

## Figures and Tables

**Figure 1 ijms-22-05713-f001:**
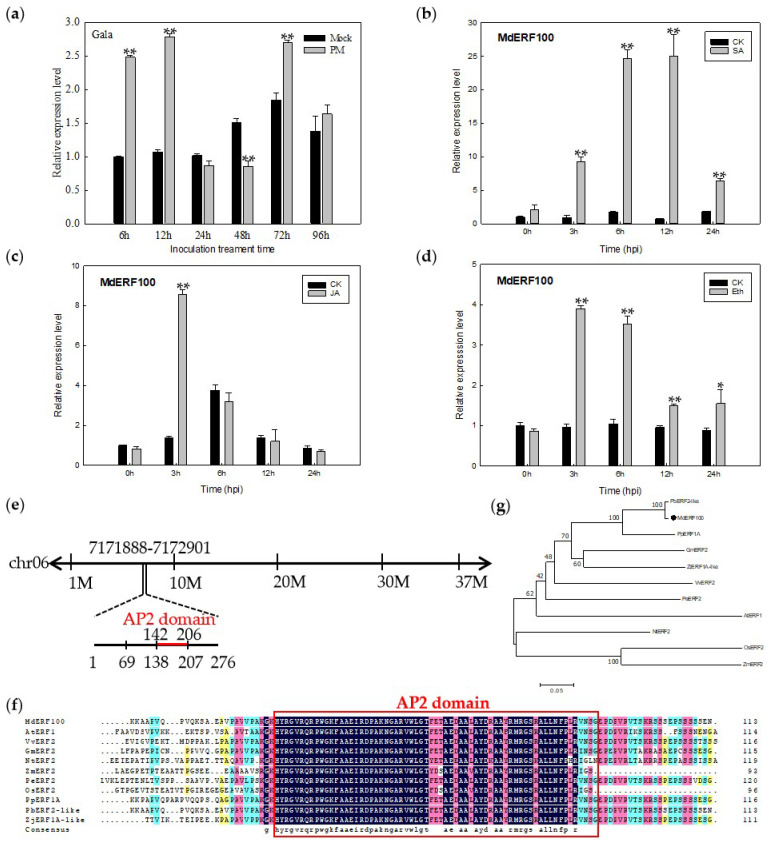
Expression and sequence analysis of *MdERF100* isolated from the apple cultivar Gala. (**a**) Quantitative real-time PCR (qRT-PCR) analysis of *MdERF100* in plants infected with powdery mildew. The x-axis indicates the time of sampling after inoculation, and the y-axis shows the relative transcription level. Error bars were calculated from three biological experiments, and show the standard deviation of the mean. Asterisks indicate statistical significance (* *p* < 0.05, ** *p* < 0.01; Student’s *t*-test). (**b**–**d**) Expression profile of *MdERF100* in leaves treated with salicylic acid (SA) (**b**), jasmonic acid (JA) (**c**) and ethephon (Eth) (**d**,**e**) Chromosomal location of *MdERF100*. (**f**) Multiple sequence alignment of MdERF100 and its homologs in 10 different species. The conserved ERF domain is outlined in red. The accession numbers of the ERFs are as follows: PbERF2-like (XP009369170.1), PpERF1A (XP007209449.1), GmERF2 (XP003538752.2), ZjERF1A-like (XP015896697.1), VvERF2 (RVW50777.1), PeERF2 (XP011031947.1), AtERF1 (NP567530.4), NtERF2 (NP001311965.1), OsERF2 (XP015623259.1), ZmERF2 (PWZ22577.1). (**g**) Phylogenetic analysis of MdERF100 and its homologs.

**Figure 2 ijms-22-05713-f002:**
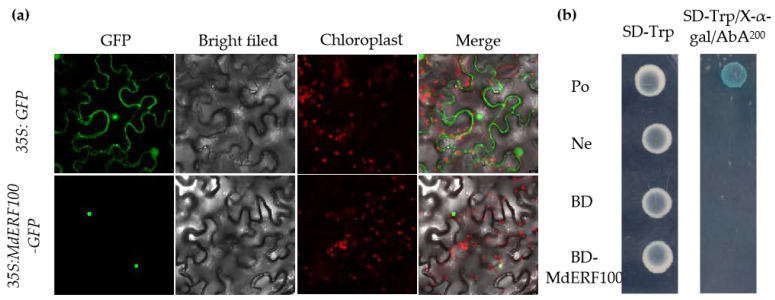
Subcellular localization and transactivation analysis of MdERF100. (**a**) Subcellular localization of MdERF100 in *Nicotiana benthamiana* plants. The pCAMBIA2300-GFP empty vector was used as control. Scale bar = 10 μm. (**b**) Transcriptional activation analysis of MdERF100 in yeast. Co-transformation of pGBKT7-53 with pGADT7-T and that of pGBKT7-Lam with pGADT7-T into yeast cells were used as positive and negative controls, respectively.

**Figure 3 ijms-22-05713-f003:**
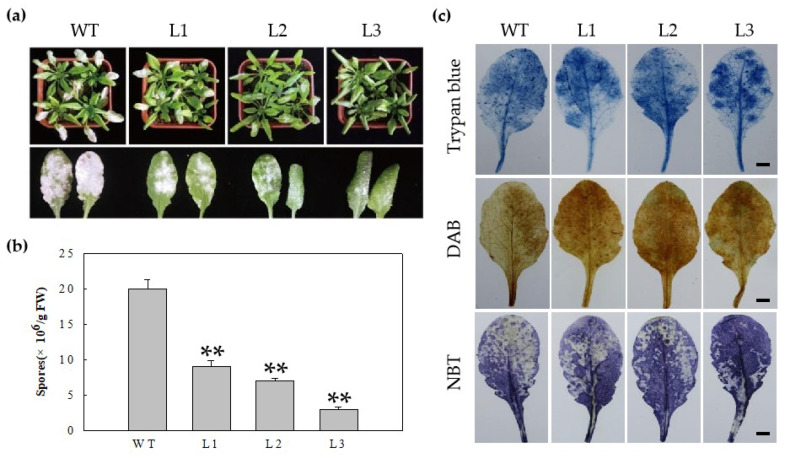
Response of *MdERF100*-overexpressing Arabidopsis lines to powdery mildew infection. (**a**) Phenotypes of wild-type (WT) plants and transgenic lines (L1, L2, L3) infected with powdery mildew. (**b**) Number of spores per gram of fresh leaves at 7 dpi (days post-inoculation). Error bars show the standard deviation of the mean. Asterisks indicate statistically significant differences (** *p <* 0.01; Student’s *t*-test). Experiments were repeated three times with consistent results. (**c**) Cell death and O^2−^ accumulation after pathogen inoculation. Scale bar = 10 mm.

**Figure 4 ijms-22-05713-f004:**
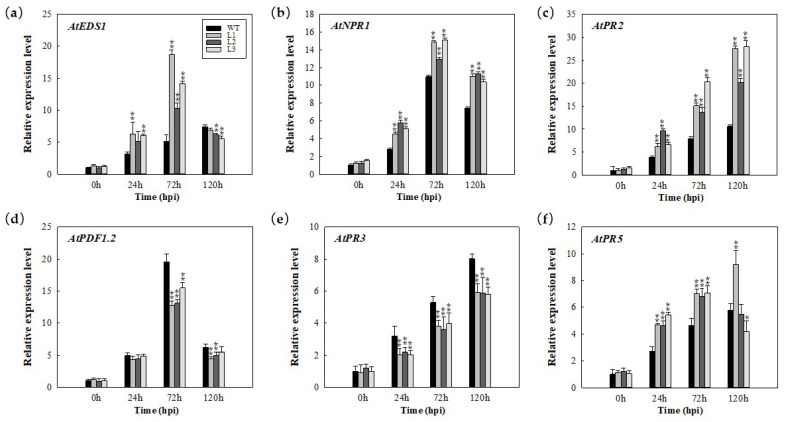
Expression analysis of disease resistance genes in transgenic *Arabidopsis*. (**a**–**f**) Expression profiles of AtEDS1 (**a**), AtNPR1 (**b**), AtPR2 (**c**), AtPDF1.2 (**d**), AtPR3 (**e**) and AtPR5 (**f**) at 0, 24, 72 and 120 hpi (hours post-inoculation). Error bars were calculated from three biological experiments, and show the standard deviation of the mean. Asterisks indicate statistical significance (* *p* < 0.05, ** *p* < 0.01; Student’s *t*-test).

**Figure 5 ijms-22-05713-f005:**
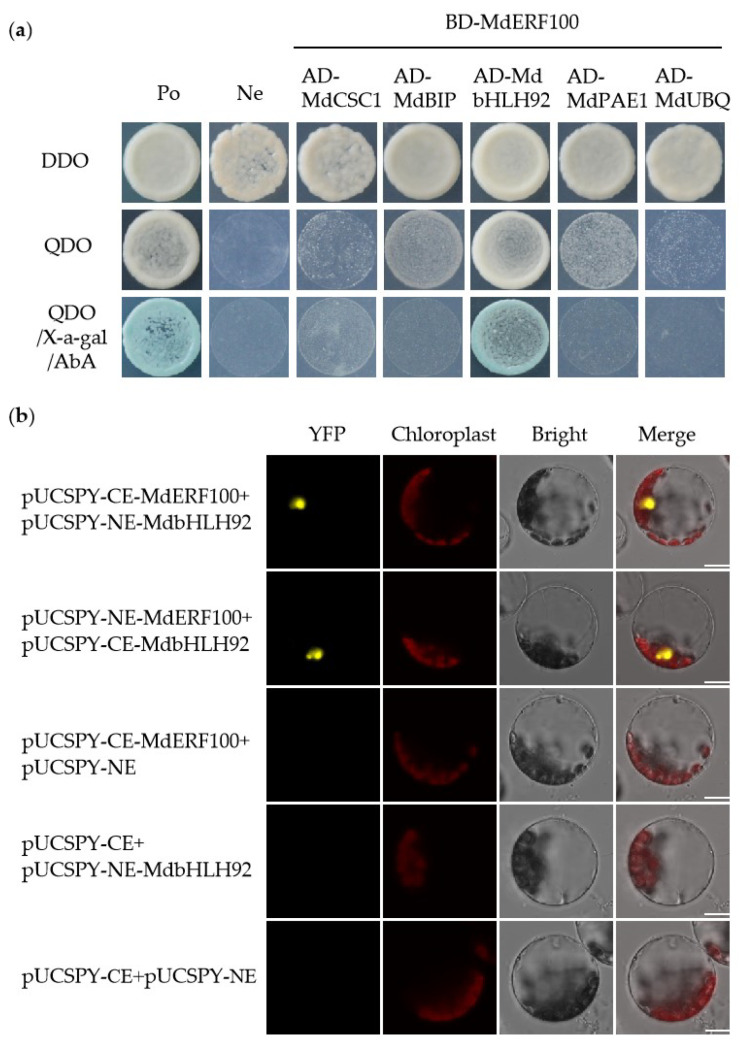
Verification of interaction between MdERF100 and MdbHLH92. (**a**) Yeast two-hybrid (Y2H) assay. The accession numbers of putative interacting partners of MdERF100 are as follows: MdCSC1 (XP_008373502.2), MdBIP (XP_008390320.1), MdPAE1 (XP_008371892.1), MdUBQ (XP_008367744.1). Yeast co-transformed with pGBKT7-53 and pGADT7-T or with pGBKT7-Lam and pGADT7-T served as a positive or negative control, respectively. (**b**) Bimolecular fluorescence complementation (BiFC) assay. Different combinations of plasmid (pUC-SPYNE-MdERF100 and pUC-SPYCE-MdbHLH92; pUC-SPYCE-MdERF100 and pUC-SPYNE-MdbHLH92; pUC-SPYCE-MdERF100 and pUC-SPYNE; pUC-SPYCE and pUC-SPYNE-MdbHLH92; pUC-SPYCE and pUC-SPYNE) were co-transformed in *Arabidopsis* protoplasts. Yellow fluorescent protein (YFP) signals were detected after 22 h. Scale bar = 10 μm.

**Figure 6 ijms-22-05713-f006:**
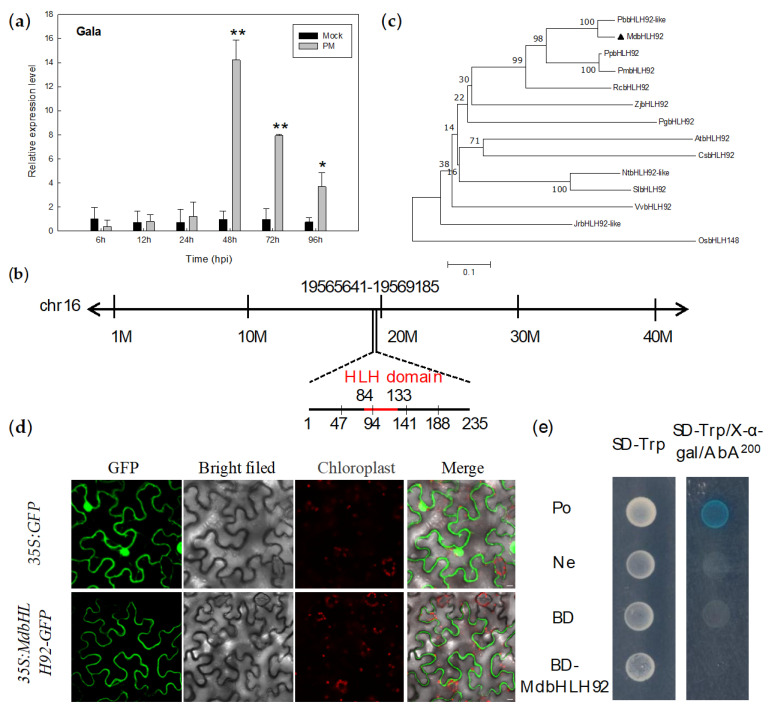
Characterization of *MdbHLH92* isolated from the apple cultivar Gala. (**a**) Expression level of *MdbHLH92* in plants infected with powdery mildew. The x-axis indicates the time of sampling after inoculation, and the y-axis shows the relative gene expression level. Asterisks indicate statistical significance (* *p* < 0.05, ** *p* < 0.01; Student’s *t*-test). Error bars were calculated from three biological experiments, and show the standard deviation of the mean. (**b**) Chromosomal location of *MdbHLH92* in the apple genome. (**c**) Phylogenetic analysis of MdbHLH92 and its homologs. The accession numbers of various proteins are as follows: JrbHLH92-like (XP_018813391.2); PbbHLH92-like (XP_009349402.1); AtbHLH92 (OAO93461.1); VvbHLH92 (XP_002284231.1); OsbHLH148 (XP_015631231.1); PpbHLH92 (XP_007227265.1); NtbHLH92-like (XP_016462348.1); PmbHLH92 (XP_016647633.1); PgbHLH92 (XP_031377233.1); ZjbHLH92 (XP_015887823.1); RcbHLH92 (XP_024194728.1); CsbHLH92 (XP_011649165.2); SlbHLH92 (XP_004247648.1). (**d**) Subcellular localization of MdbHLH92. GFP signals were observed in *Nicotiana benthamiana* plants. The pCAMBIA2300-GFP empty vector was used as a control. Scale bar = 10 μm. (**e**) Transcriptional activation analysis of MdbHLH92 in yeast. The co-transformation of pGBKT7-53 with pGADT7-T and that of pGBKT7-Lam with pGADT7-T into yeast cells were used as positive and negative controls, respectively.

**Figure 7 ijms-22-05713-f007:**
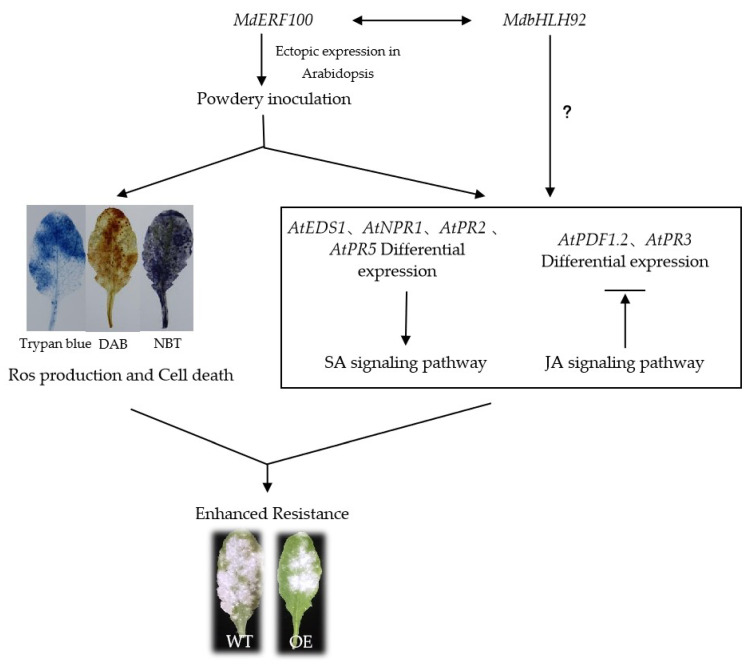
Hypothetical model depicting how MdERF100 functions as a positive regulator of responses to powdery mildew infection.

## Data Availability

The data supporting the findings of this study are available within the article and its Appendix A.

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
