# Peer review of "Overexpression of the Apple (Malus × domestica) MdERF100 in Arabidopsis Increases Resistance to Powdery Mildew"

_ijms, 2021, doi:10.3390/ijms22115713_

Round 1
Reviewer 1 Report
In this work, the researchers obtained transgenic Arabidopsis plants expressing the apple transcription factor MdERF100 to elucidate its physiological role in plant resistance to powdery mildew and its mechanism of action. Several experiments were performed to confirm that the expression of MdERF100 in Arabidopsis plants influenced the expression of genes involved in the SA and JA signaling pathways, as well as its interaction with MdbHLH92.
Major conclusions by the authors include (i) MdERF100 acts as an important regulator of protection against apple powdery mildew through signaling pathways SA, JA и Eth; (ii) fusion MdERF100-GFP localized in the nucleus; (iii) heterologous expression of MdERF100 in Arabidopsis confers powdery mildew resistance; (iv) MdERF100 positively regulates powdery mildew resistance via the SA signaling pathway, but negatively regulates powdery mildew resistance via the JA signaling pathway; (v) MdERF100 specifically interacts with MdbHLH92 in yeast cells and Arabidopsis protoplasts.
The experiments seem to be performed in sound techniques and presented in a proper manner.
The topic of this work is interesting; however, there are some comments to manuscript:
Major comments:
- Based on what experimental data or theoretical considerations was the MdERF100 gene selected from the large family of ERF transcription factor genes to elucidate its physiological role in plant resistance to powdery mildew?
- When the authors describe experimental data obtained using transgenic Arabidopsis lines expressing MdERF100 - there is no data on the level of expression of the target gene in plants. It is necessary to supplement the manuscript with such data.
- When the authors describe the results of Y2H, it would be important for the readers to know on what basis the putative MdERF100 interacting partners (MdCSC1, MdBIP, MdbHLH92, MdPAE1 и MdUBQ) were selected.
- More correctly and professionally, when using transgenic plants as experimental models for studying gene functions, use additional controls, namely, lines of transgenic plants obtained by transformation with empty vectors.
Minor comments:
- Introduction Section (Lines 72-73): it is better to use the term “defensive responses” instead of “immune responses” - Plant hormones such as SA and jasmonic acid (JA) play crucial roles in various immune responses [18]. In Arabidopsis...
- Legend to Figure 1: it is better to use the term “relative transcription level” instead of “relative gene expression level” - The x-axis indicates the time of sampling after inoculation, and y-axis shows the relative gene expression level
- When describing the key vector pCAMBIA2300-GFP used by the authors both for assessing the localization of the target gene product in a plant cell and for obtaining transgenic lines, its names according to the manufacturer should be specified and the procedure for its production should be clarified. Since the manufacturing company does not have such a vector in the line of vectors. Look at the link:CambiaLabs - https://cambia.org/welcome-to-cambialabs/cambialabs-projects/cambialabs-projects-legacy-pcambia-vectors-pcambia-legacy-vectors-1/
- When describing cloning and PCR procedures, it is more professional to indicate the number of primers, DNA or RNA templates and enzymes used, not in the volumes used, but in concentrations - ng, μg, E of activity and so on.
- For vectors used in the works, for example, pGBKT7, pUC-CE and pUC-NE, pCAMBIA2300-GFP, you should give a reference to the manufacturer or a link to a publication that describes the creation of the vectors.
Reviewer 2 Report
The paper investigates the role of MdERF100 in powdery mildew resistance in apple.
In my opinion the manuscript should be accepted after minor revision, addressing the following aspects:
The abstract is complete and describes briefly the main results and conclusions.
The introduction reports the principal pieces of information about the topic. However, it is better to provide your working hypothesis and aims of the study more clearly and precise at the end of the Introduction section.
Please, replace “Arabidopsis” with “Arabidopsis” (in italics) throughout all the manuscript.
Figure 1a, 1b, 1c, 1d should be improved. They are difficult to see.
I recommend to authors to discuss better the different stresses tested in these lines with regard to previous works because that it would improve our understanding further in the roles of MdERF100 and MdbHLH92 in both, powdery mildew resistance and plant defense against powdery mildew, respectively.
Reviewer 3 Report
In this manuscript, the Authors investigated the functional aspects of transcription factor MdERF100 from apple concerning powdery mildew disease. The present work showed the nuclear localization of MdERF100. Further, MdERF100-overexpressing Arabidopsis lines exhibited enhanced resistance to powdery mildew. Additionally, MdERF100 interaction with transcription factor MdbHLH92 provides a preliminary idea about its involvement in stress responses.
I have thoroughly read the paper. I think this manuscript is of merit for considering it for publication in IJMS. Some minor changes are needed in the text.
- Line 72-87- This background information about plant defense responses and powdery mildew studies is difficult to follow. Authors may revise the text to make it easy to follow the content.
- Figure 1e. – adding the labeling to the line of protein sequence would be helpful.
- Figure 4- Axis titles are not uniform in style and font size.
- Line 265-266- water(Figure 1a). Add space between water and (Figure 1a).
- Discussion- Heterologous expression of MdERF100 in Arabidopsis showed enhanced tolerance to powdery mildew, possibly through altered expression of defense-related genes from jasmonic acid (JA) and salicylic acid (SA) signaling pathways. To substantiate the claim- the similar role of MdERF100 in powdery mildew resistance in apple- the Authors should discuss some previous studies that shown similar aspects.
- Line 334- Six–eight-week-old- check the phrase for correctness.
Round 2
Reviewer 1 Report
After reviewing the resubmitted manuscript and the authors' responses to my comments, the following should be noted: (i) the authors satisfactorily responded to all the questions and comments I have raised; (ii) the comments and questions posed by me has been clarified in manuscript.
Author Response
Thank you for moderating the review of our manuscript, ‘Overexpression of the Apple (Malus × domestica) MdERF100 in Arabidopsis Increases Resistance to Powdery Mildew (Manuscript ID: ijms-1211893)‘.